# Eye Aspect Ratio for Real-Time Drowsiness Detection to Improve Driver Safety

**Christine Dewi** [1],*[ID]**, Rung-Ching Chen** [2],*[ID]**, Chun-Wei Chang** [2]**, Shih-Hung Wu** [3]**, Xiaoyi Jiang** [4][ID] **and Hui Yu** [5][ID]

1    Department of Information Technology, Satya Wacana Christian University, 52-60 Diponegoro Rd, Salatiga City 50711, Indonesia
2    Department of Information Management, Chaoyang University of Technology, Taichung 41349, Taiwan
3    Department of Computer Science and Information Engineering, Chaoyang University of Technology, Taichung 41349, Taiwan
4    Department of Mathematics and Computer Science, University of Münster, D-48149 Münster, Germany
5    School of Creative Technologies, University of Portsmouth, Portsmouth PO1 2UP, UK
*    Correspondence: christine.dewi@uksw.edu (C.D.); crching@cyut.edu.tw (R.-C.C.)

**Abstract:** Drowsiness is a major risk factor for road safety, contributing to serious injury, death, and economic loss on the road. Driving performance decreases because of increased drowsiness. In several different applications, such as facial movement analysis and driver safety, blink detection is an essential requirement that is used. The extremely rapid blink rate, on the other hand, makes automatic blink detection an extremely challenging task. This research paper presents a technique for identifying eye blinks in a video series recorded by a car dashboard camera in real time. The suggested technique determines the facial landmark positions for each video frame and then extracts the vertical distance between the eyelids from the facial landmark positions. The algorithm that has been proposed estimates the facial landmark positions, extracts a single scalar quantity by making use of Eye Aspect Ratio (EAR), and identifies the eye closeness in each frame. In the end, blinks are recognized by employing the modified EAR threshold value in conjunction with a pattern of EAR values in a relatively short period of time. Experimental evidence indicates that the greater the EAR threshold, the worse the AUC's accuracy and performance. Further, 0.18 was determined to be the optimum EAR threshold in our research.

**Keywords:** blink detections; driver safety; drowsiness detection; eye aspect ratio; eye blink; facial landmarks

## 1. Introduction

The technology for detecting eye blinks is important and has been used in a variety of areas, including drowsiness detection [1,2], driver safety [3–5], computer vision [6,7], and anti-spoofing protection in face recognition systems [8,9]. Drowsiness is one of the most significant variables that jeopardize road safety and contributes to serious injuries, deaths, and economic losses on the road. Due to increased drowsiness, driving performance decreases. Accidents involving serious injury or death occur because of inattention caused by an involuntary shift from waking to sleep. Individuals with normal vision exhibit spontaneous eye blinking at a specific frequency. Improvements in information and signal processing technologies have a positive impact on autonomous driving (AD), making driving safer while reducing the challenges faced by human drivers as a result of newly developed artificial intelligence (AI) techniques [10,11]. Over the decades, the development of autonomous vehicles has resulted in life-changing breakthroughs. In reality, there will be noticeable societal effects from its adoption in the areas of accessibility, safety, security, and ecology [12].

Eye blinking is influenced by various factors, including eyelid conditions, eye conditions, the presence of disease, the presence of contact lenses, psychological conditions,

the surrounding environment, drugs, and other stimuli. The number of blinks per minute ranges between 6 and 30 [13]. According to [14], the human blink rate varies depending on the circumstances. During normal activity, a person's average blink rate is 17 blinks per minute. There is variation in blink rate, with the highest being 26 blinks per minute and the lowest being 4–5 blinks per minute. In this way, it becomes clear that a person's blink rate varies based on the environment he is in and his concentration on the task at hand.

While driving, one must maintain the maximum level of concentration on the road, which results in a reduction in the blink rate. When driving, the average blinking speed is about 8–10 blinks per minute. A person's blink rate is also affected by their age group, gender, and the amount of time they spend blinking. There are real-time facial landmark detectors that can capture most of the distinctive aspects of human facial photographs. These features include the corner-of-the-eye angles and the eyelids [15,16].

The size of an individual's eye does not correspond to the size of his or her body. Take, for instance, two people who are physically identical save for the sizes of their eyes: one can have enormous eyes, and the other, small eyes. The eye height when closed is the same for all people, no matter whether the size of the eye is big or small. This problem will inevitably influence the experimental findings. In response, we present a simple but very successful method for identifying the blink of an eye using a facial landmark detector with Eye Aspect Ratio (EAR). One easy way is to use the Eye Aspect Ratio (EAR) algorithm. Further, the EAR requires only basic calculations based on the ratio of the distances between the eye's facial landmarks. This technique for detecting the blink of an eye is fast, efficient, and easy to practice. Dewi et al. [17] built their own eye dataset, which had several challenges, including small eyes, wearing glasses, and driving a car. This dataset is adapted to the characteristics of small eyes. We used this dataset in our experiment.

The model proposed in [18] is one in which the eye is modeled in conjunction with its surrounding context. The first step is a visual context pattern-based eye model, and the second step is semi-supervised boosting for high-precision detection. The approach consists of these two steps. Lee et al. [19] tried to provide an estimation of the condition of an eye, which may include whether or not an eye is open or closed.

When analyzing patterns in a visual setting, it is important to maintain as much consistency as possible in what is visible. Another approach is presented in [20], where an eye filter is utilized for finding all the eye candidate's points. The non-negative matrix factorization (NMF) is reduced to its smallest possible value because of this reconstruction of the mistake.

Pioneering work was performed by Fan Li et al. [21], who investigated the effects of data quality on eye-tracking-based fatigue indicators and proposed a hierarchical-based interpolation approach to extract eye-tracking-based fatigue indicators from low-quality eye-tracking data. This work was considered groundbreaking because it investigated the effects of data quality on eye-tracking-based fatigue indicators. Gracia et al. [22] conducted an experiment using eye closure for separate frames, which was then subsequently used in a sequence for the detection of blinks. We built our algorithm upon the successful methods of Eye Aspect Ratio [23,24] and facial landmarks [25,26].

Learning and normalized cross-correlation are used to create templates with open and/or closed eyes. Eye blinks can also be identified by measuring ocular parameters, such as by fitting ellipses to eye pupils using a variation of the algebraic distance algorithm for conic approximation. Eye blinks can be detected by monitoring ocular parameters [27,28].

The following is a list of the most significant contributions that this article has made: (1) We propose a method to automatically classify blink types by determining the different EAR thresholds (0.18, 0.2, 0.225, 0.25). (2) Adjustments were made to the Eye Aspect Ratio to improve the detection of eye blinking based on facial landmarks. (3) We conducted an in-depth analysis of the experiment's findings using TalkingFace, Eyeblink8, and Eye Blink datasets. (4) Our experimental results show that using 0.18 as the EAR threshold provides the best performance.

The following is the structure of this research work. The Materials and Methods section covers related work and the methodology we applied in this research. Results and Discussion describes our experimental setting and results. In the final section, conclusions are drawn and suggestions for future research are made.

## 2. Materials and Methods

Drowsiness is characterized by yawning, heavy eyelids, daydreaming, eye rubbing, an inability to concentrate, and lack of attention. The percentage of eyelid closure over the pupil over time (PERCLOS) [29] is one of the most widely used parameters in computer-vision-based drowsiness detection in driving scenarios [30]. In reference [31], a convolutional neural network (CNN) was used to develop a tiredness detection system. The program trained the first network to distinguish between human and non-human eyes, then used the second network to locate the eye feature points and calculate the eye-opening degree. Efficient algorithms for detecting drowsiness are presented in this article. In addition, facial landmarks can be retrieved using the Dlib toolkit. By identifying the different EAR thresholds, we present a method for automatically classifying blink types.

### 2.1. Facial Landmarks for Eye Blink Detection

Deep-learning-based facial landmark detection systems have made impressive strides in recent years [32,33]. A cascaded convolutional network model, as proposed by Sun et al. [34], consists of a total of 23 CNN models. This model has very high computational complexity during training and testing.

To detect and track important facial features, identification of facial markers must be performed on the subject. As a result of head movements and facial expressions, facial tracking is stronger for rigid facial deformations. Facial landmark identification is a computer vision job in which we try to identify and track key points on the human face using computer vision algorithms [35]. Multi-Block Color-Binarized Statistical Image Features (MB-C-BSIF) is discussed in [36]; this is a novel approach to single-sample facial recognition (SSFR) that makes use of local, regional, global, and textured-color properties [37,38].

Drowsiness can be measured on a computational eyeglass that can continually sense fine-grained measures such as blink duration and percentage of eye closure (PERCLOS) at high frame rates of about 100 fps. This work can be used for a variety of problems. Facial landmarks are used to localize and represent salient regions of the face, including eyes, eyebrows, nose, mouth, and jawline.

Blinking occurs repeatedly and involuntarily throughout the day to maintain a certain thickness of the tear film on the cornea [39]. The act of blinking is a reflex that involves the fast closure and opening of the eyelids in rapid succession. Blinking is also known as blepharospasm. The act of blinking is performed subconsciously. The synchronization of several different muscles is required for the act of blinking one's eyes.

While keeping the cornea healthy is an important function of blinking, there are other benefits as well [40], and this is supported by the fact that adults and infants blink their eyes at different rates. A person's blink frequency changes in response to their level of activity. The number of blinks increases when a person reads a certain phrase aloud or performs a visually given information exercise, whereas the number of blinks decreases when a person focuses on visual information or reads words quietly [41].

In our investigation, we made use of the 68 facial landmarks from Dlib [42]. Estimating the 68 (x,y)-coordinates corresponding to the facial structure on the face was carried out with the help of a pre-trained facial landmark detector found in the Dlib library. Figure 1 displays that the jaw points range from 0 to 16, the right brow points range from 17 to 21, and the left brow points range from 22 to 26. The nose points range from 27 to 35, the right eye points range from 36 to 41, and the left eye points range from 42 to 47. The mouth points range from 48 to 60, and the lip points range from 61 to 67. Dlib is a library that helps implement computer vision and machine learning techniques. The C++ programming language serves as the foundation for this library.

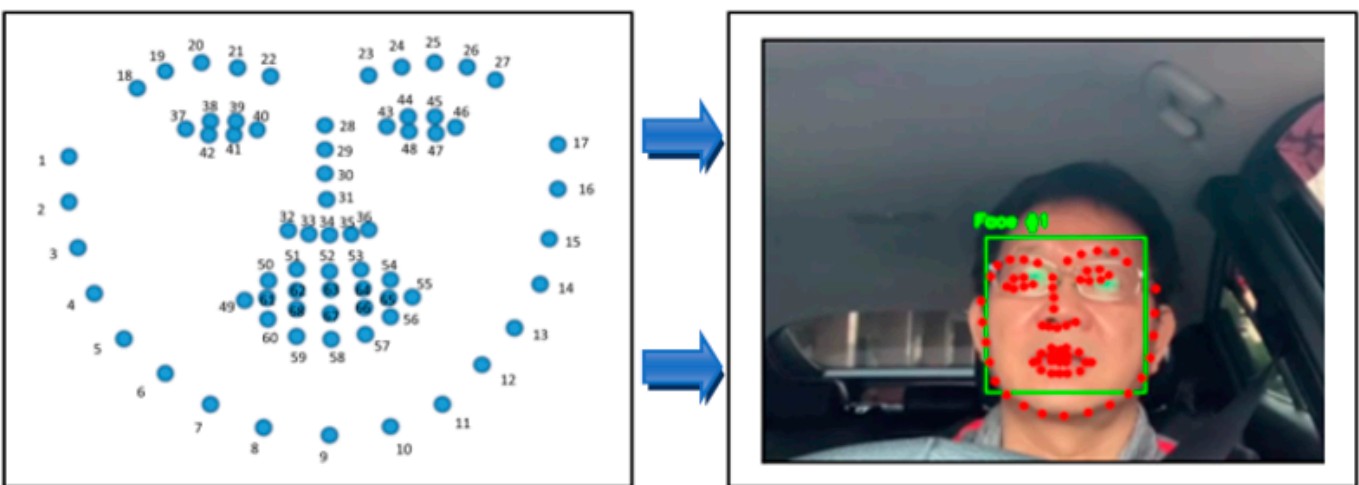

**Figure 1.** Eye identification made possible by employing facial landmarks (right eye points = 36–41, left eye points = 42–47).

The process of locating facial landmark points with the use of Dlib's 68-landmark model consists of two stages, all of which are described in the following order: (1) The first way to locate a human face, face detection, is done by returning a value in the form of x, y, w, and h coordinates, which together form a rectangle. (2) A facial landmark: Once we have determined the location of a face within an image, we must then place points within the rectangle. This annotation is included in the 68-point iBUG 300-W dataset, which serves as the basis for the training of the Dlib facial landmark predictor. The Dlib framework can be utilized to train form predictors on input training data, regardless of the dataset that is selected to be trained on.

*2.2. Eye Aspect Ratio (EAR)*

The Eye Aspect Ratio, or EAR, is a scalar value that responds, particularly for opening and closing the eyes [43]. During the flashing process, we can see that the EAR value grows rapidly or decreases significantly. Interesting findings in terms of robustness were obtained when EAR was used to detect blinks in [44]. Studies in the past have employed a predetermined EAR threshold to establish when subjects blink (EAR threshold at 0.2). This approach is impractical when dealing with a wide range of individuals, due to inter-subject variation in appearance and features such as natural eye openness, as in this study. Our works used an EAR threshold value to detect a rapid increase or decrease in the EAR value caused by blinking, based on the findings of previous studies.

We used the varying EAR threshold to automatically categorize the various sorts of blinks (0.18, 0.2, 0.225, 0.25). After that, we analyzed the experimental result and determined the best EAR threshold for our dataset. Each frame of the video stream is used to estimate the EAR. Furthermore, when the user shuts their eyes, the EAR drops and then returns to a regular level when the eyes are opened again. This technique is used to determine both blinks and eye opening. As the EAR formula is insensitive to both the direction of the face and the distance between it and the observer, it can be used to detect faces from a considerable distance. The EAR value can be calculated by entering six coordinates surrounding the eyes, as shown in Figure 2, and Equations (1) and (2) [30,45].

$$EAR = \frac{\| P2 - P6 \| + \| P3 - P5 \|}{2 \| P1 - P4 \|} \tag{1}$$

$$AVG\ EAR = \frac{1}{2}(EAR_{Left} + EAR_{Right}) \tag{2}$$

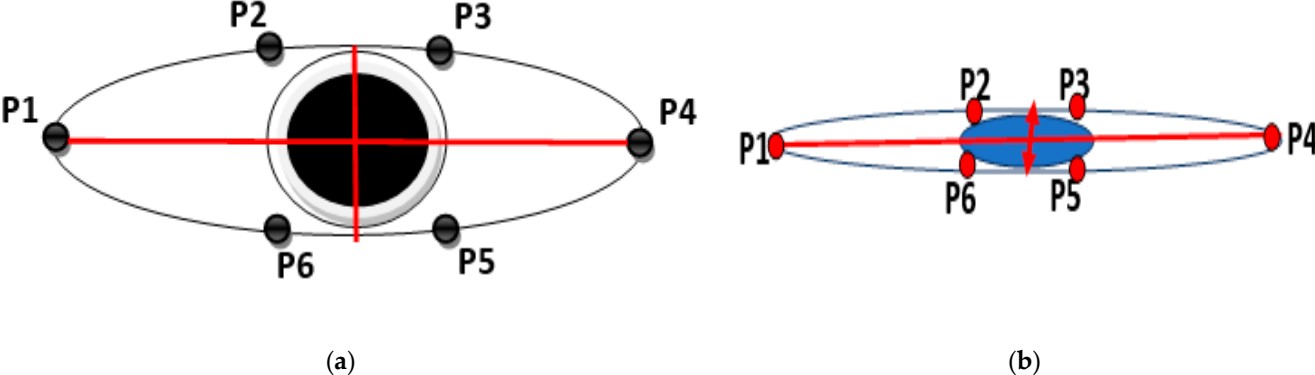

**Figure 2.** Open and closed eyes with facial landmarks (P1, P2, P3, P4, P5, P6). (**a**) Open eye. (**b**) Close eye.

The EAR equations are described by Equation (1), where P1 through P6 stand in for the locations of the 2D landmarks on the retina. P2, P3, P5, and P6 were utilized to measure the height of the eye, whereas P1 and P4 were utilized to measure the breadth of the eye. This is depicted in Figure 2. When the eyes are closed, the EAR value quickly drops to virtually zero, in contrast to when the eyes are open, in which case the EAR value remains constant. This behavior is seen in Figure 2b.

*2.3. Research Workflow*

Our system architecture is divided into two steps, namely, data preprocessing and eye blink detection, as described in Figure 4. In the data preprocessing step, the video labeling procedure using Eyeblink Annotator 3.0 by Andrej Fogelton [46] is shown in Figure 3. OpenCV version 2.4.6 is utilized by the annotation tool. Both video 1 and video 3 were recorded at a frame rate of 27.97 frames per second. Video 2 was captured with 24 fps. Video 1 has a length of 1829 frames, totaling 29.6 MB. Video 2 has a length of 1302 frames and a file size of 12.4 MB. Next, video 3 has 2195 frames and a file size of 38.6 MB. The Talking Face and Eyeblink8 datasets contain 5000 frames and 10,712 frames, respectively. Video information is explained in Table 1.

People who wear glasses and have relatively tiny eyes are represented in our dataset in a unique way. The people who operate automobiles make up the environment. This dataset may be utilized for additional research endeavors. Based on what we know, it is difficult to locate a dataset of persons who have tiny eyes, wear spectacles, and drive cars. We have the footage from the dashboard camera installed in a vehicle in the Wufeng District of Taichung, Taiwan. We have verified that informed consent was received from each individual who participated in the video dataset collection. Our data collection includes five films and one individual performing a driving scenario.

The annotations start with line "#start," and rows consist of the following information: frame ID: blink ID: NF: LE_FC: LE_NV: RE_FC: RE_NV: F_X: F_Y: F_W: F_H: LE_LX: LE_LY: LE_RX: LE_RY: RE_LX: RE_LY: RE_RX: RE_RY. An example of a frame with a blink is: 118: −1: X: X: X: X: X: 394: 287: 220: 217: 442: 369: 468: 367: 516: 364: 546: 363. The eyes may or may not be completely closed during a blink. According to the website blinkmatters.com, the range of fully closed eyes during a blink is between ninety percent to one hundred percent [46]. The row will be like: 415: 5: X: C: X: C: X: 451: 294: 182: 191: 491: 362: 513: 363: 554: 365: 577: 367. In this particular study, our experiments were only interested in the blink ID and eye completely closed (FC) columns; as a result, we ignored any additional information that may be provided. Table 2 provides an explanation of the features included in the dataset.

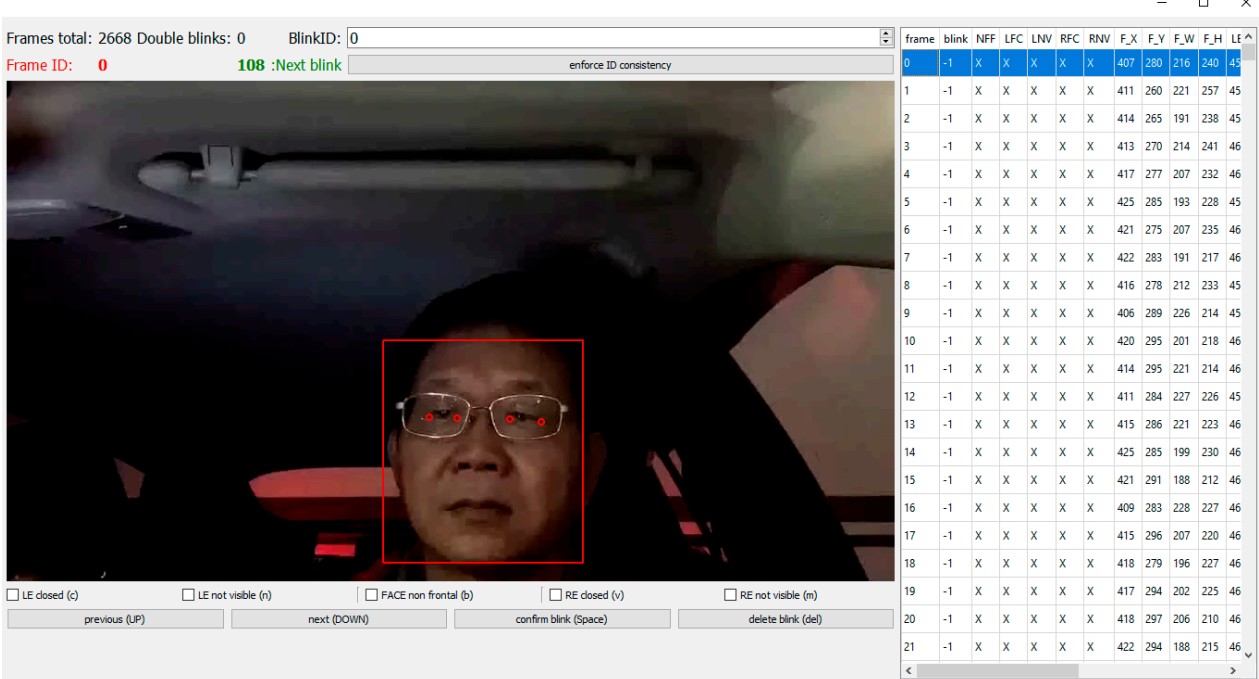

**Figure 3.** Video labeling process with Eyeblink Annotator 3.0.

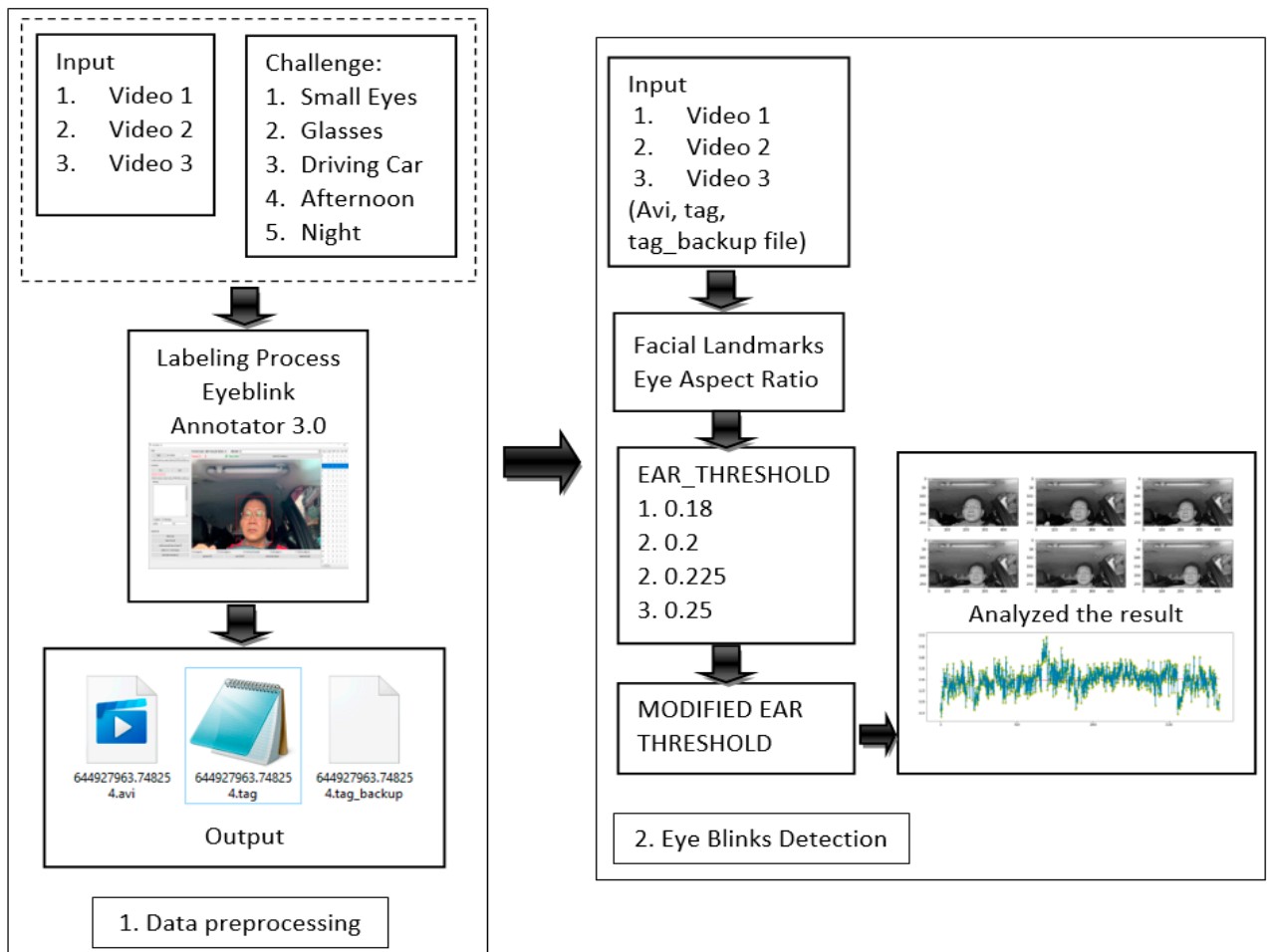

**Figure 4.** The system architecture.

**Table 1.** Video dataset information.

| Video Info | Video 1 | Video 2 | Video 3 | Talking Face | Eyeblink8 Video 8 |
|---|---|---|---|---|---|
| FPS | 29.97 | 24 | 29.97 | 30 | 30 |
| Frame Count | 1829 | 1302 | 2195 | 5000 | 10,712 |
| Durations (s) | 61.03 | 54.25 | 73.24 | 166.67 | 357.07 |
| Size (MB) | 29.6 | 12.4 | 38.6 | 22 | 18.6 |

**Table 2.** Dataset features.

| No | Description | Features |
|---|---|---|
| 1 | Alternatively, a frame counter may be used to get a timestamp in a different file. | frame ID |
| 2 | A unique blink ID is defined as a sequence of blink ID frames that are all the identical. The time between two consecutive blinks is measured in terms of a sequence of identical blink ID frames. | blink ID |
| 3 | A change from X to N occurs in the provided variable while the person is looking sideways and blinking. | non frontal face (NF) |
| 4 | Left Eye. | left eye (LE), |
| 5 | Right Eye. | right eye (RE), |
| 6 | Face. | face (F) |
| 7 | The given flag will transition from X to C if the subject's eye closure percentage is between 90% and 100%. | eye fully closed (FC) |
| 8 | This variable changes from X to N when the subject's eye is covered (by the subject's hand, by low lighting, or by the subject's excessive head movement). | eye not visible (NV) |
| 9 | x and y coordinates, width, height. | face bounding box (F_X, F_Y, F_W, F_H) |
| 10 | RX (right corner x coordinate), LY (left corner y coordinate) | left and right eye corners positions |

The Eyeblink8 dataset is more complex because it includes facial expressions, head gestures, and staring down at a keyboard. According to [46], this dataset has a total of 408 blinks across 70,992 video frames at $640 \times 480$-pixel resolution. This clip has an average length of between 5000 and 11,000 frames and was shot at 30 frames per second. There is only one video of a single participant chatting to the camera, so to speak, in the Talking Face dataset. In the video, the person can be seen smiling, laughing, and doing a "funny face" in a variety of situations. In addition, the frame rate is 30 frames per second, the resolution is 720 by 576, and 61 blinks that have been labeled.

*2.4. Eye Blink Detection Flowchart*

Figure 5 illustrates the blink detection method, and the frame-by-frame breakdown of the video is the initial stage. The facial landmarks feature [47] was implemented with the help of Dlib to detect the face. The detector used here is made up of classic histogram of oriented gradients (HOG) [48] feature along with a linear classifier. In order to identify face characteristics, including the ears, eyes, and nose, a facial landmarks detector was built inside Dlib [49,50]. Moreover, with the help of two lines, our research was able to identify blinks. The lines dividing the eye are drawn in two directions: horizontal and vertical. Blinking is the act of briefly closing the eyes and shifting the eyelids from one side to the other. Blinking is a natural thing to happen.

The eyes are closed or blinking when the eyeballs are not visible, the eyelids are closed, the upper and lower eyelids are fused, and the upper and lower eyelids are not connected. Further, when the eyes are opened, the vertical and horizontal lines are almost the same, but the vertical lines narrow or almost disappear when the eyes are closed. We may consider eye blinking if the EAR is less than the modified EAR threshold for three seconds. To perform our experiment, we used four alternative threshold values: 0.18, 0.2, 0.225, and 0.25. Additionally, we experimented with different EAR cutoffs and both video datasets.

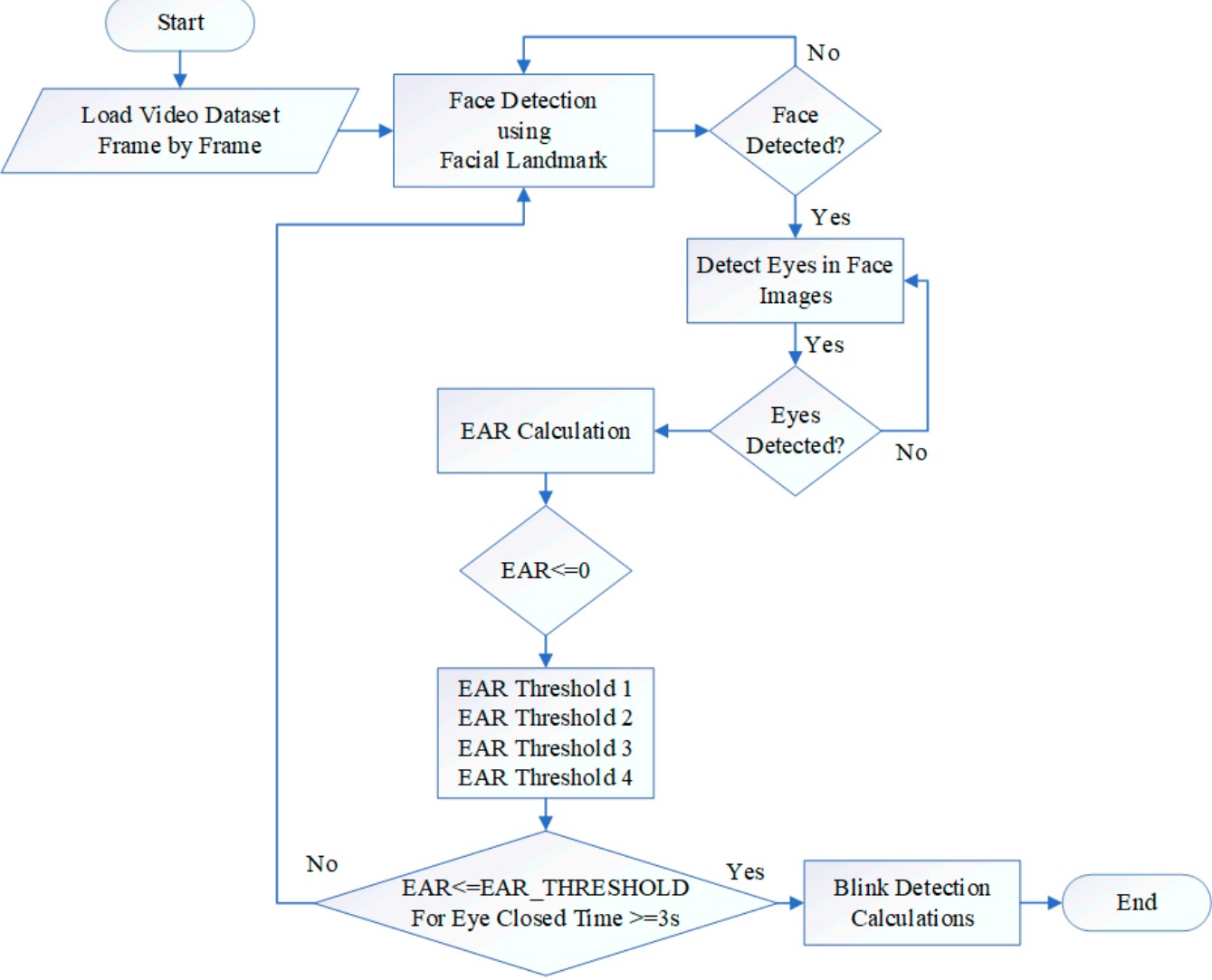

**Figure 5.** Eye blink detection flowchart.

## 3. Results

Table 3 summarizes the statistics for the predictions and test sets of video 1, video 2, and video 3. For the EAR threshold of 0.18, the total frame count for the prediction set of videos 1 is 1829, the number of closed frames analyzed was 23, and the number of blinks found was 2. On the other hand, the statistics for the test set state that there are 58 closed frames, and there are 14 blinks. This experiment exhibited an accuracy of 95.5% and an area under the curve (AUC) of 0.613. Furthermore, video 3 has 1302 frames with 182 closed frames. The maximum accuracy was obtained while implementing the 0.18 EAR threshold: 86.1%. Moreover, video 3 contains 2192 frames, totaling 73.24 seconds. Using an EAR threshold of 0.18 resulted in 89% accuracy for this dataset. In the third video, we see the minimal results of 47.5% accuracy and 0.594 AUC for an EAR threshold of 0.25 being used.

Moreover, Table 4 describes the statistics for the prediction and test sets of Talking Face and Eyeblink8 datasets. Talking Face has 5000 frames with 227 closed frames. The optimum accuracy was achieved while employing the 0.18 EAR threshold: 97.1% accuracy and 0.974 AUC. For Eyeblink8, video 8, the highest accuracy was obtained when using the 0.18 EAR threshold: 86.1% accuracy and 0.732 AUC. This dataset has 1302 frames with 182 closed frames and 18 blinks.

**Table 3.** Statistics for prediction and test sets (Eye Blink Dataset).

| Dataset | Video 1 | | | | Video 2 | | | | Video 3 | | | |
|---|---|---|---|---|---|---|---|---|---|---|---|---|
| EAR Threshold (t) | 0.18 | 0.2 | 0.225 | 0.25 | 0.18 | 0.2 | 0.225 | 0.25 | 0.18 | 0.2 | 0.225 | 0.25 |
| Statistics on the prediction set are | | | | | | | | | | | | |
| Total Number of Frames Processed | 1829 | 1829 | 1829 | 1829 | 1302 | 1302 | 1302 | 1302 | 2192 | 2192 | 2192 | 2192 |
| Number of Closed Frames | 23 | 56 | 131 | 281 | 182 | 342 | 614 | 884 | 232 | 440 | 791 | 1177 |
| Number of Blinks | 2 | 6 | 9 | 16 | 18 | 39 | 73 | 65 | 25 | 49 | 79 | 89 |
| Statistics on the test set are | | | | | | | | | | | | |
| Total Number of Frames Processed | 1829 | 1829 | 1829 | 1829 | 1302 | 1302 | 1302 | 1302 | 2192 | 2192 | 2192 | 2192 |
| Number of Closed Frames | 58 | 58 | 58 | 58 | 35 | 35 | 35 | 35 | 61 | 61 | 61 | 61 |
| Number of Blinks | 14 | 14 | 14 | 14 | 9 | 9 | 9 | 9 | 10 | 10 | 10 | 10 |
| Eye Closeness Frame by Frame Test Scores | | | | | | | | | | | | |
| Accuracy | 0.955 | 0.938 | 0.897 | 0.820 | 0.861 | 0.74 | 0.543 | 0.340 | 0.890 | 0.797 | 0.645 | 0.475 |
| AUC | 0.613 | 0.581 | 0.528 | 0.501 | 0.732 | 0.692 | 0.654 | 0.591 | 0.664 | 0.641 | 0.626 | 0.594 |

**Table 4.** Statistics on prediction and test (Talking Face and Eyeblink8 datasets).

| Dataset | Talking Face | | | | Eyeblink8 Video 8 | | | |
|---|---|---|---|---|---|---|---|---|
| EAR Threshold (t) | 0.18 | 0.2 | 0.225 | 0.25 | 0.18 | 0.2 | 0.225 | 0.25 |
| Statistics on the prediction set are | | | | | | | | |
| Total Number of Frames Processed | 5000 | 5000 | 5000 | 5000 | 10,663 | 10,663 | 10,663 | 10,663 |
| Number of Closed Frames | 227 | 292 | 352 | 484 | 404 | 529 | 1055 | 2002 |
| Number of Blinks | 31 | 42 | 49 | 59 | 37 | 43 | 85 | 126 |
| Statistics on the test set are | | | | | | | | |
| Total Number of Frames Processed | 5000 | 5000 | 5000 | 5000 | 10,663 | 10,663 | 10,663 | 10,663 |
| Number of Closed Frames | 153 | 153 | 153 | 153 | 107 | 107 | 107 | 107 |
| Number of Blinks | 61 | 61 | 61 | 61 | 30 | 30 | 30 | 30 |
| Eye Closeness Frame by Frame Test Scores | | | | | | | | |
| Accuracy | 0.971 | 0.968 | 0.959 | 0.933 | 0.970 | 0.959 | 0.911 | 0.911 |
| AUC | 0.974 | 0.968 | 0.953 | 0.946 | 0.963 | 0.961 | 0.955 | 0.955 |

The best EAR threshold in our experiment was 0.18. This value provided the best accuracy and AUC values in all experiments. Hence, 0.25 is the worst EAR threshold value because it obtained the minimum accuracy and AUC values. Based on the experimental results, it can be concluded that the higher the EAR threshold, the lower the accuracy and AUC performance. In previous studies, it was said that the EAR threshold of 0.2 is the best value, but it was not for our experiment. In fact, 0.18 was the best EAR threshold in our work. Our dataset is unique because of the small eyes. The size of the eyes will certainly affect the EAR and EAR threshold values. Therefore, our dataset also has some challenges, namely, people driving cars and people wearing glasses.

Figure 6 shows the confusion matrix for the video 1, video 2, and video 3. Figure 6a describes the false positive (FP) value of 58 out of 58 positive labels (1.0000%) and false negative (FN) rate of 23 out of 1771 negative labels (0.0130%) for video 1 and EAR threshold 0.18. Figure 6c explains the false positive (FP) rate of 35 out of 61 positive labels (0.5738%) and false negative (FN) rate of 206 out of 2131 negative labels (0.0967%).

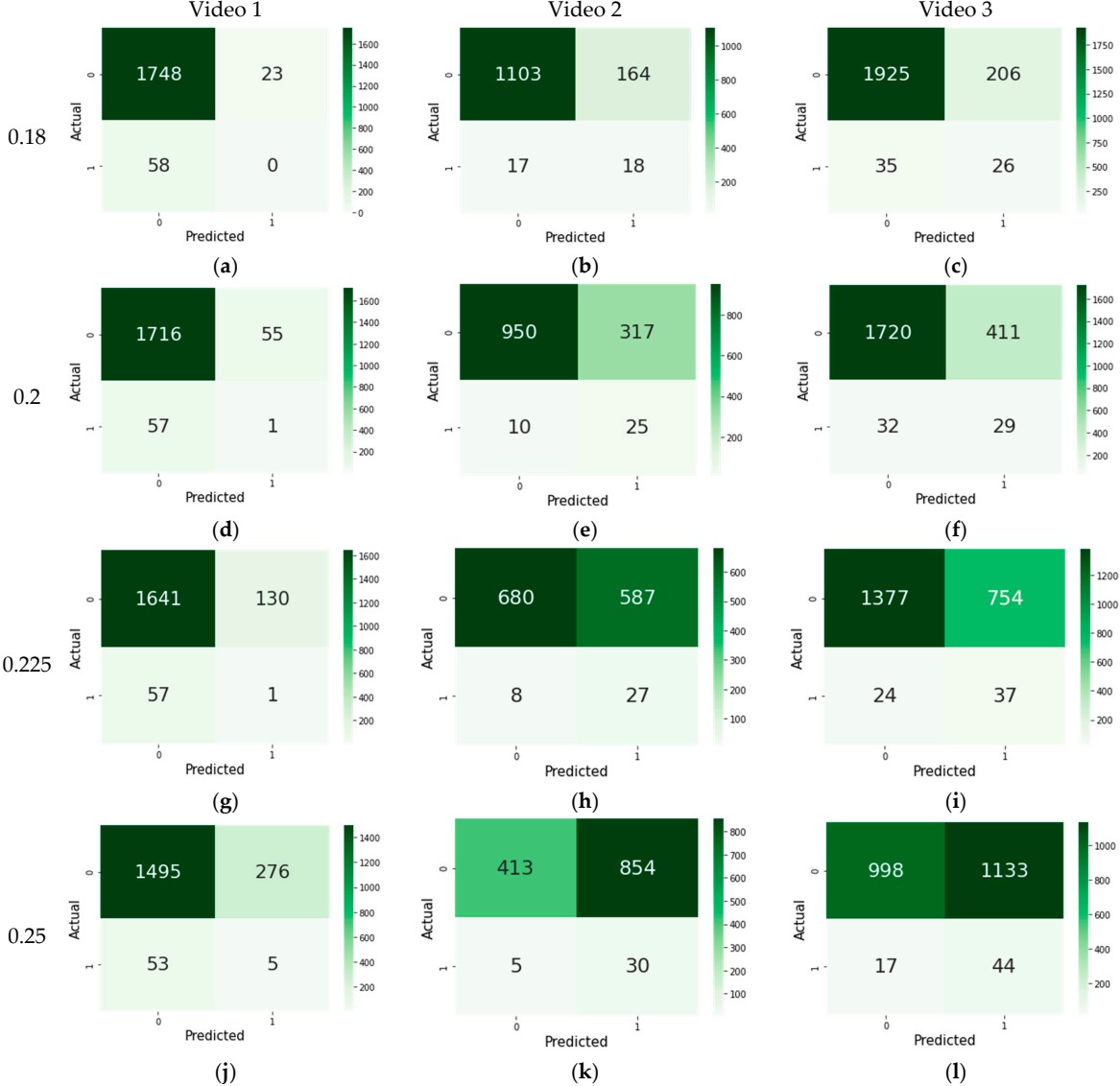

**Figure 6.** Confusion matrix (Eye Blink dataset). (**a**) Video 1 with 0.18 EAR threshold, (**b**) Video 2 with 0.18 EAR threshold, (**c**) Video 3 with 0.18 EAR threshold, (**d**) Video 1 with 0.2 EAR threshold, (**e**) Video 2 with 0.2 EAR threshold, (**f**) Video 3 with 0.2 EAR threshold, (**g**) Video 1 with 0.225 EAR threshold, (**h**) Video 2 with 0.225 EAR threshold, (**i**) Video 3 with 0.225 EAR threshold, (**j**) Video 1 with 0.25 EAR threshold, (**k**) Video 2 with 0.25 EAR threshold, (**l**) Video 3 with 0.25 EAR threshold.

In our experiment, we analyzed videos frame by frame and identified eye blinks every three frames, as shown in Figure 7. The results of the experiment only show the blinks at the beginning, middle, and end frames. For instance, Figure 7a illustrates the 1st blink started in the 3rd frame, the middle of the action was in the 5th frame, and it ended in the 7th frame. Next, Figure 7b describes that the 2nd blink started in the 1555th frame, the middle of action was in the 1556th frame, and it ended in the 1557th frame. Moreover, Figure 7c explains the 3rd blink started in the 1563rd frame, the middle of the action was in the 1564th frame, and it ended in the 1565th frame.

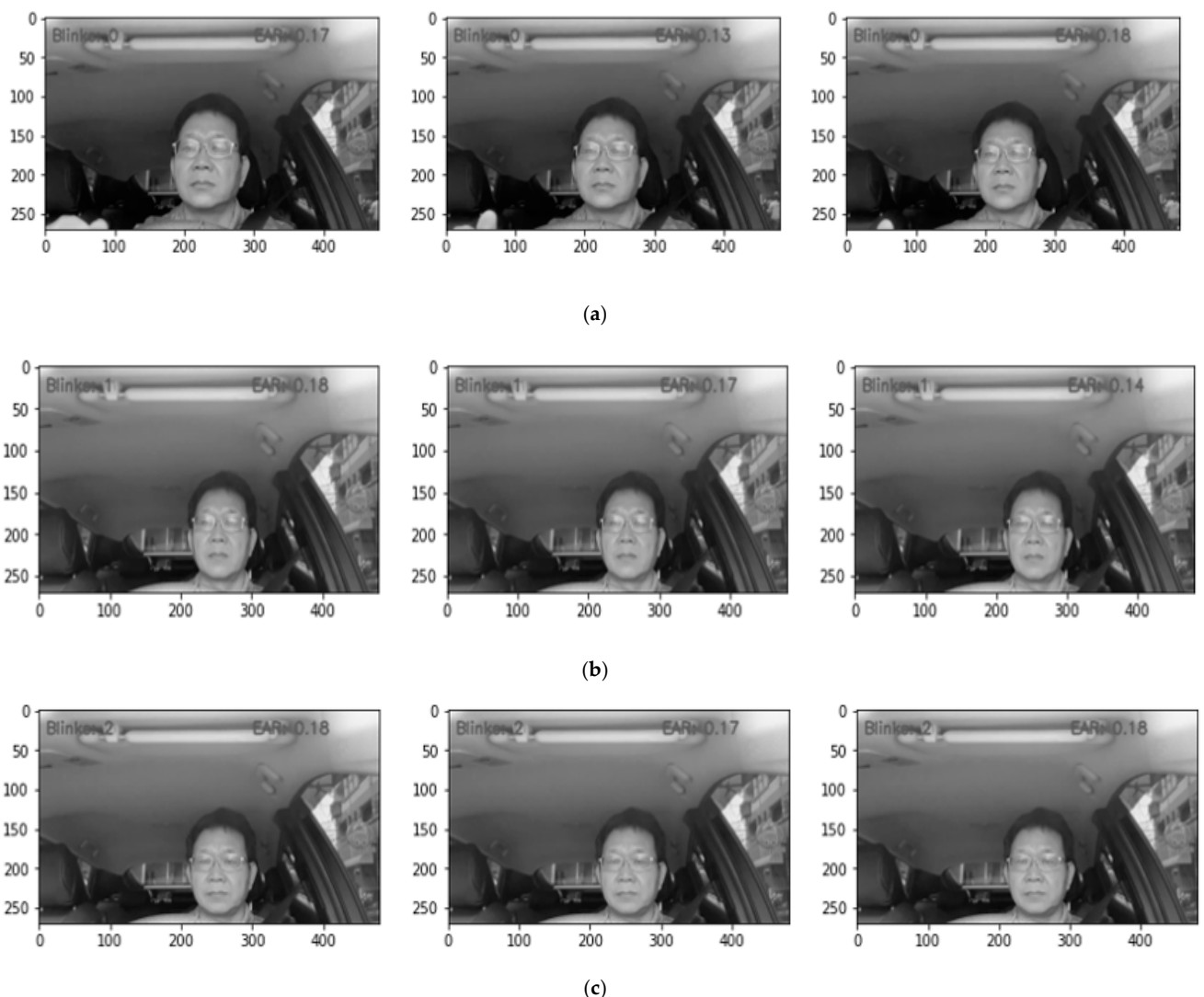

**Figure 7.** Video 1 eye blink prediction results frame by frame (threshold = 0.18). (**a**) The 1st blink started in the 3rd frame, the middle of the action in the 5th frame, and it ended in the 7th frame. (**b**) The 2nd blink started in the 1555th frame, the middle of the action was in the 1556th frame, and it ended in the 1557th frame. (**c**) The 3rd blink started in the 1563rd frame, the middle of the action was in the 1564th frame, and it ended in the 1565th frame.

Figure 8 exhibits the Video 3 eye blink prediction result frame by frame with the EAR threshold of 0.18. The 1st blink started in the 69th frame, the middle of the action was in the 71st frame, and it ended in the 72nd frame as shown in Figure 8a. The 2nd blink started in the 227th frame, the middle of the action was in the 230th frame, and it ended in the 232nd frame, as described in Figure 8b. Next, Figure 8c illustrates that the 3rd blink started in the 272nd frame, the middle of the action was in the 273rd frame, and it ended in the 274th frame.

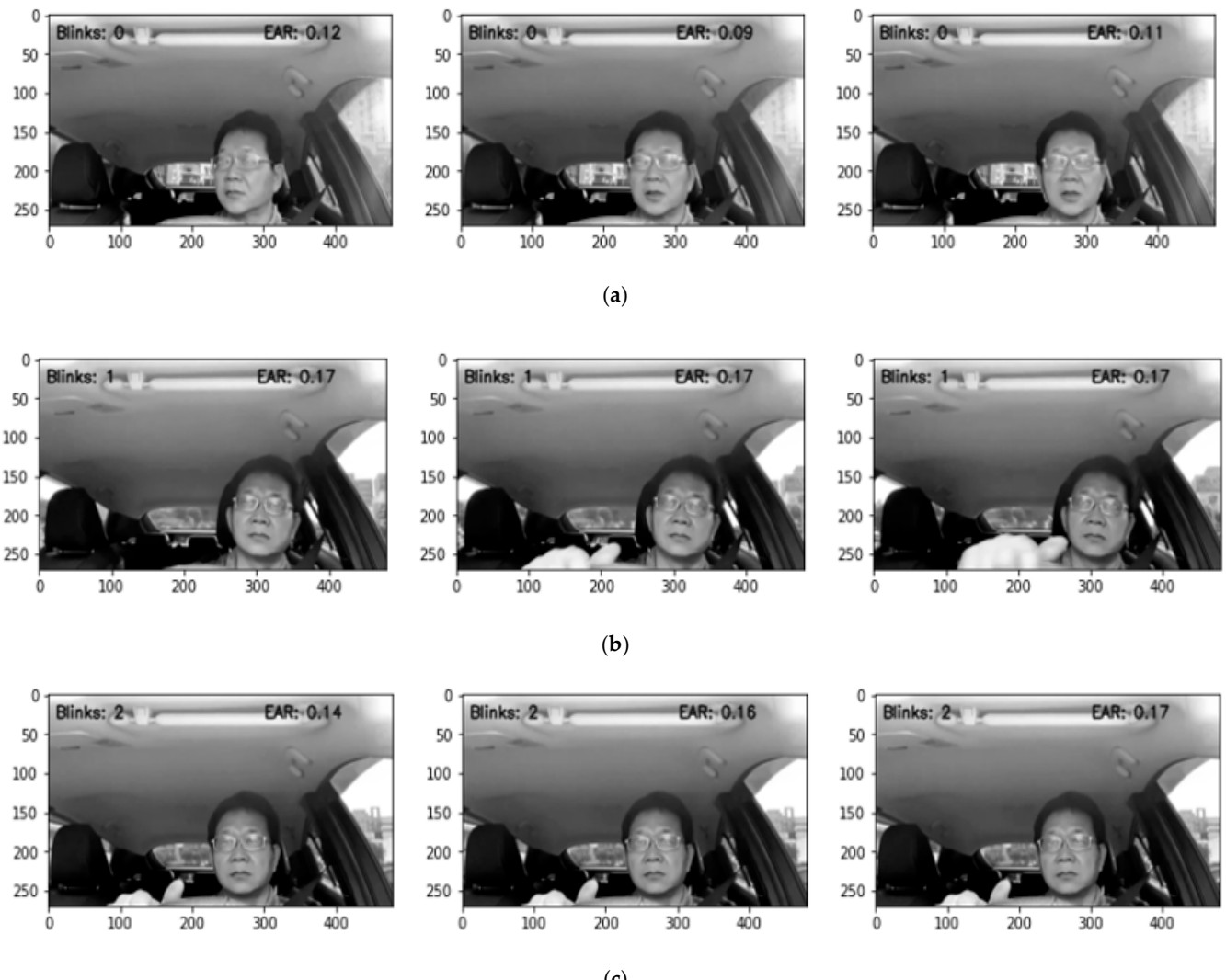

**Figure 8.** Video 2 Eye Blink Prediction result frame by frame (Threshold = 0.18). (**a**) The 1st blink started at the 69th frame, the middle of the action was in the 71st frame, and it ended in the 72nd frame. (**b**) The 2nd blink started at the 227th frame, the middle of the action was in the 230th frame, and it ended in the frame. (**c**) The 3rd blink started at the 272tnd frame, the middle of the action was in the 273rd frame, and it ended in the 274th frame.

Tables 5 and 6 present the specific results that were obtained for each video dataset. These tables include the precision, recall, and F1-score measurements. In our testing, the best EAR threshold was 0.18, which was excellent. In all tests, this number yielded the highest accuracy and AUC values, respectively. As mentioned, 0.25 was the worst EAR threshold value, since it only achieved the bare minimum in terms of accuracy and AUC.

Researchers normally choose 0.2 or 0.3 as the EAR threshold, despite the fact that not everyone's eye size is the same. As a result, it is better to recalculate the EAR threshold to detect whether the eye is closed or open in order to identify the blinks more accurately. For video 1 we achieved 96% accuracy, followed by video 3, with 89% accuracy, and video 2, with 86% accuracy, as shown in Table 5. Using Talking Face and Eyeblink8 datasets, we obtained the same accuracy, 97%, by employing the 0.18 EAR threshold, as shown in Table 6.

**Table 5.** Precision, recall, and F1-score (Eye Blink dataset).

| Evaluation | Video 1 | | | | Video 2 | | | | Video 3 | | | |
|---|---|---|---|---|---|---|---|---|---|---|---|---|
| | Precision | Recall | F1-Score | Support | Precision | Recall | F1-Score | Support | Precision | Recall | F1-Score | Support |
| | EAR Threshold (t) = 0.18 | | | | EAR Threshold (t) = 0.18 | | | | EAR Threshold (t) = 0.18 | | | |
| 0 | 0.97 | 0.99 | 0.98 | 1771 | 0.98 | 0.87 | 0.92 | 1267 | 0.98 | 0.90 | 0.94 | 2131 |
| 1 | 0.00 | 0.00 | 0.00 | 58 | 0.10 | 0.51 | 0.17 | 35 | 0.11 | 0.43 | 0.18 | 61 |
| Macro avg | 0.48 | 0.49 | 0.49 | 1829 | 0.54 | 0.69 | 0.55 | 1302 | 0.55 | 0.66 | 0.56 | 2192 |
| Weight avg | 0.94 | 0.96 | 0.95 | 1829 | 0.96 | 0.86 | 0.90 | 1302 | 0.96 | 0.89 | 0.92 | 2192 |
| Accuracy | | | 0.96 | 1829 | | | 0.86 | 1302 | | | 0.89 | 2192 |
| | EAR Threshold (t) = 0.2 | | | | EAR Threshold (t) = 0.2 | | | | EAR Threshold (t) = 0.2 | | | |
| 0 | 0.97 | 0.97 | 0.97 | 1771 | 0.99 | 0.75 | 0.85 | 1267 | 0.98 | 0.81 | 0.89 | 2131 |
| 1 | 0.02 | 0.02 | 0.02 | 58 | 0.07 | 0.71 | 0.13 | 35 | 0.07 | 0.48 | 0.12 | 61 |
| Macro avg | 0.49 | 0.49 | 0.49 | 1829 | 0.53 | 0.73 | 0.49 | 1302 | 0.52 | 0.64 | 0.50 | 2192 |
| Weight avg | 0.94 | 0.94 | 0.94 | 1829 | 0.96 | 0.75 | 0.83 | 1302 | 0.96 | 0.80 | 0.86 | 2192 |
| Accuracy | | | 0.94 | 1829 | | | 0.75 | 1302 | | | 0.80 | 2192 |
| | EAR Threshold (t) = 0.225 | | | | EAR Threshold (t) = 0.225 | | | | EAR Threshold (t) = 0.225 | | | |
| 0 | 0.97 | 0.93 | 0.95 | 1771 | 0.99 | 0.54 | 0.70 | 1267 | 0.98 | 0.65 | 0.78 | 2131 |
| 1 | 0.01 | 0.02 | 0.01 | 58 | 0.04 | 0.77 | 0.08 | 35 | 0.05 | 0.61 | 0.09 | 61 |
| Macro avg | 0.49 | 0.47 | 0.48 | 1829 | 0.52 | 0.65 | 0.39 | 1302 | 0.51 | 0.63 | 0.43 | 2192 |
| Weight avg | 0.94 | 0.90 | 0.92 | 1829 | 0.96 | 0.54 | 0.68 | 1302 | 0.96 | 0.65 | 0.76 | 2192 |
| Accuracy | | | 0.90 | 1829 | | | 0.54 | 1302 | | | 0.65 | 2192 |
| | EAR Threshold (t) = 0.25 | | | | EAR Threshold (t) = 0.25 | | | | EAR Threshold (t) = 0.25 | | | |
| 0 | 0.97 | 0.84 | 0.90 | 1771 | 0.99 | 0.33 | 0.49 | 1267 | 0.98 | 0.47 | 0.63 | 2131 |
| 1 | 0.02 | 0.09 | 0.03 | 58 | 0.03 | 0.85 | 0.07 | 35 | 0.04 | 0.72 | 0.07 | 61 |
| Macro avg | 0.49 | 0.47 | 0.47 | 1829 | 0.51 | 0.59 | 0.28 | 1302 | 0.51 | 0.59 | 0.35 | 2192 |
| Weight avg | 0.94 | 0.82 | 0.87 | 1829 | 0.96 | 0.34 | 0.48 | 1302 | 0.96 | 0.48 | 0.62 | 2192 |
| Accuracy | | | 0.82 | 1829 | | | 0.34 | 1302 | | | 0.48 | 2192 |

**Table 6.** Precision, recall, and F1-score (Talking Face and Eyeblink8 datasets).

| Evaluation | Talking Face | | | | Eyeblink8 Video 8 | | | |
|---|---|---|---|---|---|---|---|---|
| | Precision | Recall | F1-Score | Support | Precision | Recall | F1-Score | Support |
| | EAR Threshold (t) = 0.18 | | | | EAR Threshold (t) = 0.18 | | | |
| 0 | 0.99 | 0.98 | 0.99 | 4847 | 1.00 | 0.97 | 0.98 | 10,556 |
| 1 | 0.52 | 0.77 | 0.62 | 153 | 0.24 | 0.92 | 0.38 | 107 |
| Macro avg | 0.76 | 0.87 | 0.80 | 5000 | 0.62 | 0.94 | 0.68 | 10,663 |
| Weight avg | 0.98 | 0.97 | 0.97 | 5000 | 0.99 | 0.97 | 0.98 | 10,663 |
| Accuracy | | | 0.97 | 5000 | | | 0.97 | 10,663 |
| | EAR Threshold (t) = 0.2 | | | | EAR Threshold (t) = 0.2 | | | |
| 0 | 1.00 | 0.97 | 0.98 | 4847 | 1.00 | 0.96 | 0.98 | 10,556 |
| 1 | 0.49 | 0.93 | 0.64 | 153 | 0.19 | 0.96 | 0.32 | 107 |
| Macro avg | 0.74 | 0.95 | 0.81 | 5000 | 0.60 | 0.96 | 0.65 | 10,663 |
| Weight avg | 0.98 | 0.97 | 0.97 | 5000 | 0.99 | 0.96 | 0.97 | 10,663 |
| Accuracy | | | 0.97 | 5000 | | | 0.96 | 10,663 |
| | EAR Threshold (t) = 0.225 | | | | EAR Threshold (t) = 0.225 | | | |
| 0 | 1.00 | 0.96 | 0.98 | 4847 | 1.00 | 0.91 | 0.95 | 10,556 |
| 1 | 0.43 | 0.99 | 0.60 | 153 | 0.10 | 1.00 | 0.18 | 107 |
| Macro avg | 0.71 | 0.97 | 0.79 | 5000 | 0.55 | 0.96 | 0.57 | 10,663 |
| Weight avg | 0.98 | 0.96 | 0.97 | 5000 | 0.99 | 0.91 | 0.95 | 10,663 |
| Accuracy | | | 0.96 | 5000 | | | 0.91 | 10,663 |
| | EAR Threshold (t) = 0.25 | | | | EAR Threshold (t) = 0.25 | | | |
| 0 | 1.00 | 0.93 | 0.96 | 4847 | 1.00 | 0.91 | 0.95 | 10,556 |
| 1 | 0.32 | 1.00 | 0.48 | 153 | 0.10 | 1.00 | 0.18 | 107 |
| Macro avg | 0.66 | 0.97 | 0.72 | 5000 | 0.55 | 0.96 | 0.57 | 10,663 |
| Weight avg | 0.98 | 0.93 | 0.95 | 5000 | 0.99 | 0.91 | 0.95 | 10,663 |
| Accuracy | | | 0.93 | 5000 | | | 0.91 | 10,663 |

## 4. Discussion

The EAR and error analysis of the video 1 dataset is presented in Figure 9. The EAR threshold for this dataset was set to 0.18. Assume that the linear regression's optimum slope is m ≥ 0. All the data from video 1 were plotted in our experiment, and the result is m = 0. Infrequent blinking has a small impact on the overall trend in EAR measurements depicted in Figure 10a. However, the cumulative error is irrelevant for blinks, owing to its delayed effect. Nevertheless, mistakes behave more similarly to correctly dispersed data than the EAR values in Figure 10b.

The effectiveness of the proposed method for detecting eye blinks was evaluated in this work by contrasting the detected blinks with the ground-truth blinks using the three video datasets. Overall, the output samples may be divided into three categories. The samples that were correctly identified are referred to as true positives (TP), the samples that had the wrong identification are referred to as false positives (FP), and the samples that were correctly not recognized are referred to as true negatives (TN) [51,52]. Precision (P) and recall (R) are represented by [53,54] in Equations (3) and (4).

$$P = \frac{TP}{TP + FP} \tag{3}$$

$$R = \frac{TP}{TP + FN} \tag{4}$$

Another evaluation index, F1 [55], is shown in Equation (5).

$$F1 = \frac{2 \times \text{Precision} \times \text{Recall}}{\text{Precision} + \text{Recall}} \tag{5}$$

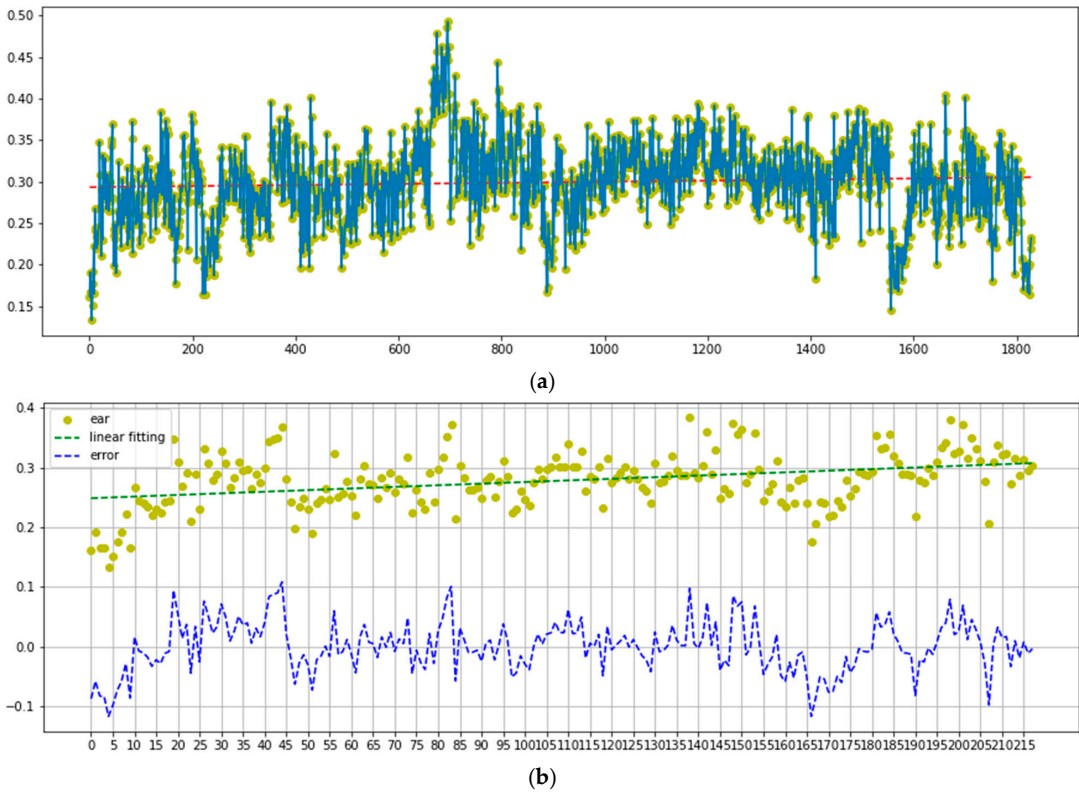

(**a**)

(**b**)

**Figure 9.** EAR and error analysis (Video 1, EAR threshold = 0.18). (**a**) Average EAR. (**b**) Error and EAR.

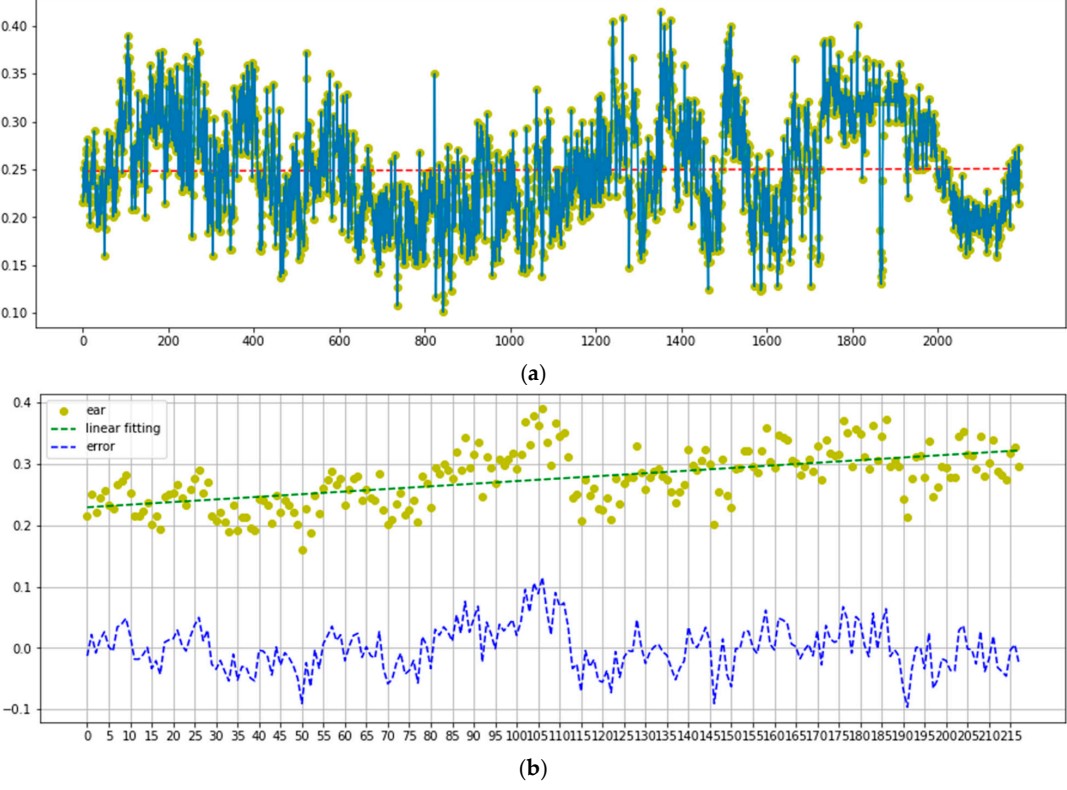

(**a**)

(**b**)

**Figure 10.** EAR and error analysis (Video 3, EAR threshold = 0.18). (**a**) Average EAR. (**b**) Error and EAR.

There is a possibility that if the driver does not blink for a long time and his EAR value decreases without any blinking in the initial period, the algorithm will not return an error. Further, our work calculates errors as *errors = calibration − linear* and *cumsum ()*. This function will return the cumulative sum of the elements along a given axis. A new array holding the result, in which case a reference to out is returned, is returned unless out is specified. The result has the same size and shape as if the axis were none or a 1d array. Cumulative errors are not very important for blinking, as their effects are delayed. However, typical errors can be exploited to detect anomalies. Figure 10 describes the EAR and error analysis of the video 3 dataset with an EAR threshold of 0.18. The average EAR value for video 3 was 0.25, as shown in Figure 9b. This average value is slightly different from the average value in Figure 10a, which is close to 0.30. During our tests, we found that the optimal EAR threshold is 0.18. This figure yielded the greatest accuracy and AUC values in all tests, both excellent results. The statistics are listed in Tables 5 and 6.

Furthermore, Table 7 describes the evaluation of the proposed method in comparison to existing research. Our proposed method achieved peak average accuracies of 97.10% with the TalkingFace dataset, 97.00% with the Eyeblink8 dataset, and 96% with the Eye Video 1 dataset. We improved on the performances of previous methods.

**Table 7.** Evaluation of the proposed method in comparison to existing research.

| Reference | Dataset Accuracy (%) | | |
|---|---|---|---|
| | Talking Face | Eyeblink8 | Eye Video 1 |
| Drutarovskys et al. [7] | 92.20 | 79.00 | - |
| Fogelton et al. [46] | 95.00 | 94.69 | - |
| Proposed Method | 97.10 | 97.00 | 96.00 |

## 5. Conclusions

In this article, we provide a method for automatically classifying blink types which includes establishing a threshold based on the Eye Aspect Ratio value. We call it Real-Time Blink Detection for Driver Safety using Eye Aspect Ratio. Using our eye blink dataset, we conducted a thorough analysis of the method. According to the experimental findings, the higher the EAR threshold, the worse the accuracy and AUC. Previously published studies showed that an EAR threshold of 0.2 was the optimal value; however, this was not the case in our experiment. In our study, 0.18 was the optimal EAR threshold. The experimental findings suggest that the EAR threshold for identifying whether the eyes are open or closed should be recalculated. Machine-learning techniques could be a viable alternative. In our future research, we will use explainable artificial intelligence (XAI) to explain our model when making certain predictions, as this is as important as prediction accuracy. In addition, our future work will explore the use of generative adversarial networks (GANs) to generate new synthetic data samples and improve image representation or quality [56,57]. We may consider coloring an image in grayscale, enhancing coloring, denoising, segmenting, or removing occlusion by objects [58].

**Author Contributions:** Conceptualization, C.D., R.-C.C. and X.J.; data curation, C.D. and S.-H.W.; formal analysis, C.D., S.-H.W. and H.Y.; investigation, C.D. and C.-W.C.; methodology, C.D.; project administration, R.-C.C. and X.J.; resources, C.D. and C.-W.C.; software, C.D., C.-W.C. and X.J.; supervision, R.-C.C., S.-H.W. and H.Y.; visualization, H.Y.; writing—original draft, C.D.; writing—review and editing, C.D. All authors have read and agreed to the published version of the manuscript.

**Funding:** This paper is supported by the Ministry of Science and Technology, Taiwan. The numbers are MOST-111-2221-E-324 -020, and MOST-110 -2927-I-324 -501- Taiwan. Additionally, this study was partially funded by the EU Horizon 2020 program RISE Project ULTRACEPT under grant 778062.

**Institutional Review Board Statement:** Ethical review and approval were waived for this study due to all subjects gave their informed consent for inclusion before they participated in the study.

**Informed Consent Statement:** All subjects gave their informed consent for inclusion before they participated in the study.

**Data Availability Statement:** Taiwan Eye Blink Dataset (https://drive.google.com/drive/u/1/folders/1U2qlw-ViqdW1pny77aJGLUIEf0B-HKzZ (accessed on 13 January 2021)).

**Acknowledgments:** The authors would like to thank the support and help from Chaoyang University of Technology, Satya Wacana Christian University, and others that took part in this work.

**Conflicts of Interest:** The authors declare no conflict of interest.

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
