# Peer review of "Eye Aspect Ratio for Real-Time Drowsiness Detection to Improve Driver Safety"

_electronics, doi:10.3390/electronics11193183_

Round 1

Reviewer 1 Report

This paper proposes an eye blink detection method in a video series to reduce the driving drowsiness risk. The method classifies blink types by determining the different EAR thresholds and adjustment is made to the eye aspect ratio to improve the detection of eye blinking based on facial landmarks.

Generally, technological innovation is not obvious in this paper. Blink detection is not a new topic. All key modules in the proposed architecture are direct adoptions of existing methods.  The combination of these methods to detect blinking is also not particularly special. It is more like a project development based on existing methods rather than scientific research.

The purpose or motivation of this paper is not quite clear. There is a lot of blink detection research. What is the specialty of this paper to make an improvement or difference? Or, does this paper resolves any problem of existing methods?

Moreover, in the experiment part, the method of this paper is not compared with any other recent blink detection work. It is hard to evaluate its performance without comparisons.

In the title and abstract, the method is described as a real-time drowsiness detection method. The character of "real-time" is very important for drowsiness detection, yet there is no corresponding experiment or analysis to demonstrate it.

Also, there are some typos, such as 1th, 2th, 3th in figure 7.

Reviewer 2 Report

In spite of there is low scientific novelty, authors presented their papaer well. 

Author Response

Dear Chief Editor, 

Many thanks for allowing us to revise our manuscript for possible publication in the Journal of Electronics. The paper is titled " Eye Aspect Ratio for Real-Time Drowsiness Detection to Improve Driver Safety”. We have modified the manuscript accordingly, and detailed corrections are listed below point by point:

Comments:

Reviewer #2:

  1. In spite of there is low scientific novelty, authors presented their paper well. 

Responses:

Thanks to the reviewer for the comment.

Reviewer 3 Report

The study is an interesting one. The authors presented a technique for 19 identifying eye blinks in a video series recorded by a car dashboard camera in real-time. If the authors formulate some research questions and provide their answers in the discussion/conclusion section, that would be better. Also, the knowledge gap should be presented. The limitation of the study should be highlighted. A comparison of the study with the state-of-the-art studies in this domain may be incorporated. Higher the AUC value higher the accuracy. The authors achieved an accuracy of 95.5% with an AUC value of 0.613(line 266). Are the results correct?

Round 2

Reviewer 1 Report

The authors have modified the manuscript and answered the questions. I have no further question.